



**Estimating changes of temperatures and precipitation extremes in India using the**
**Generalized Extreme Value (GEV) distribution**

**Kishore Pangaluru[1][*], Isabella Velicogna[1,2], Tyler C. Sutterley[1], Yara Mohajerani[1],**
**Enrico Ciracì[1], Jyothi Sompalli[3], and Saranga Vijaya Bhaskara Rao[3]**
1. Department of Earth System Science, University of California, Irvine, California,
92697, USA
2. Jet Propulsion Laboratory, California Institute of Technology, Pasadena, California,
USA
3. Department of Physics, Sri Venkateswara University, Tirupati, India
**\*Corresponding Author:**
**Email: kishore@uci.edu; Ph:1-949-824-3516**



**Abstract**

Changes in extreme temperature and precipitation may give some of the largest

significant societal and ecological impacts. For changes in the magnitude of extreme
temperature and precipitation over India, we used a statistical model of generalized
extreme value (GEV) distribution. The GEV statistical distribution is a time-dependent
distribution with different time scales of variability bounded by a precipitation, maximum
($T_{max}$), and minimum ($T_{min}$) temperature extremes and also assessed their possibility
changes are evaluated and quantified over India is presented. The GEV-based method is
applied on both precipitation and temperature extremes over India during the 20[th] and 21[st]
centuries using multiple coupled climate models taking an interest in the Coupled Model
Intercomparison Project Phase 5 (CMIP5) and observational datasets. The regional means
of historical warm extreme temperatures are 34.89, 36.42, and 38.14 $^{o}$C for three different
(10, 20, and 50-year) periods, respectively; whereas the cold extreme mean temperatures
are 7.75, 4.19, and -1.57 $^{o}$C. It indicates that 20th century cold extreme temperatures have
relatively larger variations than the warm extremes. As for the future, the CMIP5 models
of warm extreme regional mean values increase from 0.33 to 0.75 $^{o}$C in all return periods
(10-, 20-, and 50-year periods), while in the case of cold extreme means values vary
between 0.58 and 2.29 $^{o}$C. In the future, cold extreme values have a larger increasing rate
over the northwest, northeast, some parts of north central, and Inter Peninsula regions.
The CRU precipitation extremes are larger than the historical extreme precipitation in all
three (10, 20, and 50-year) return-periods.

Keywords: Precipitation, surface temperature, GEV, Historical, and CMIP5.



## 1. Introduction


Extreme weather events, amplified by climate change, can lead to major
environmental issues affecting human society. Precipitation and temperature are two
major components of a changing climate that have been analyzed extensively over the
past two decades (Trenberth and Shea 2005; Li et al., 2009; Kharin et al., 2013).
According to the United Nations Office for Disaster Risk Reduction UNISDR (2015),
India is the third most influenced nation by weather related by disasters, which can
largely be attributed to both higher occurrences of extreme temperatures and precipitation.
Recently, Trenberth (2005) showed that climate change due to increased greenhouse gas
emissions leads to changes in extreme event behavior in terms of precipitation and
temperature all over the world. Generalized Extreme Value (GEV) statistical distribution
has long been used to examine time-series of climate extremes with different return levels
using three extreme value distributions that were proposed by Fisher and Tippet (1928).
The three distributions are referred to as Gumbel, Frechet, and negative Weibull, which
are discussed in Section 2. Jaruskova and Rencova (2008) studied the extreme changes in
annual maxima and minima temperature series using five meteorological sites,
implementing extreme value theory and hypothesis testing within the framework of the
GEV-based method.
Jenkinson (1955) used GEV distribution for extreme precipitation events, which
offered extensive adaptability of the three extreme value distributions. Later, several
researchers used GEV statistical distribution to study extreme precipitation for many
regions and different countries around the world (Fowler and Kilsby 2003; Nadarajah
2005; and Gilleland and Katz 2006). In China, a warming trend has been confirmed in



both annual minimum and maximum temperature in the twentieth century (Choi et al.
2009; You et al. 2011). Later studies also showed notable extreme temperature increases
in northeastern China, and the smallest increase in the southern region (Liu et al. 2004).
The frequency of extreme temperature events in China is expected to increase at an
accelerating rate based on Coupled Model Inter-comparison Project (CMIP) historical
projections (Wang and Chen 2014; Yang et al. 2014). Utilization of GEV distribution on
temperature and precipitation over China has been extensively studied in several
investigations (Wang and Zhou 2005; Zhang et al. 2011; Yang et al. 2014). As for India,
Shashikanth et al. (2017) applied a GEV distribution to GCM summer monsoon
precipitation in India during 1961-1990 and 2081-2100. They found a slight increase in
the future extreme spatial mean in the later period. However, the statistical GEV
distributions of extreme minimum and maximum temperatures in India have not been
examined in any previous studies. We utilize this method over India to address this issue.
CMIP models and observations are discussed in Section 2. The GEV statistical
distribution methodology is described in Section 3. Section 4 presents the results of the
GEV distribution in three different periods and occurrences over India, and finally the
conclusions are discussed in Section 5.
**2. Data and Method**
The observational dataset of gridded monthly precipitation (P), minimum and
maximum surface temperatures ($T_{min}$ and $T_{max}$) are taken from the study of the Climate
Research Unit (CRU TS3.1) described by Harris et al. (2014). The datasets are collected
from 1901 to 2005 over land areas, based on daily values from rain gauge measurements
provided by more than 4,000 weather stations distributed around the world (New et al.,



1999, 2000). The precipitation and surface temperatures are collected from different
sources, with rigorous quality checking procedures before gridding (Mitchell and Jones,
2005; Harish et al., 2014). Figure 1 shows the Indian map with seven regions.

The monthly precipitation, and the minimum and maximum surface temperatures

($T_{min}$ and $T_{max}$) are simulated by CMIP5 (Coupled Intercomparison Project Phase 5)
models for a historical (hereafter referred to as "Historical") period from 1850 to 2005
(Smith et al., 2013; Lamarque et al., 2010) as well as the 21st century (years 2006-2100)
employing four different representative concentration pathways (RCPs) (Moss et al.,
2010, Taylor et al., 2012). The Historical and different scenarios of CMIP5 models are
listed in Table 1. Further details on the models and their configuration are described in
the references, online at http://www-pcmdi.llnl.gov/. We have considered only models for
which the same ensemble member i.e. 'r1i1p1' is available both in the historical and four
(RCP2.6, RCP4.5, RCP6.0, and RCP8.5) scenarios considered here. According to the
IPCC Fifth Assessment Report, the CMIP5 models exhibit improvements in the
simulations especially surface temperature and precipitation compared to the previous
climate models (Flato et al. 2013). The outputs for both historic and different RCPs
outputs are available on different spatial scales, which are consequently regridded to a
common spatial scale of 1$^{o}$ in latitude and 1$^{o}$ in longitude resolution.

Out of the monthly CMIP5 model outputs (listed in Table 1), Historical

experiments, RCP (2.6, 4.5, 6.0, and 8.5) experiments of $T_{min}$, $T_{max}$, and Precipitation (P)
are utilized for our analysis.





Three types of extreme distributions compose a GEV distribution: Gumbel,

Frèchet, and Weibull, also known as type I, II, and III respectively (Martins and
Stedinger 2000; Feng et al., 2007). These can generally be described by

$$G((z; \mu, \sigma, \xi) = \begin{cases} exp\left\{-exp\left[-\left(\frac{z-\mu}{\sigma}\right)\right]\right\}, & \xi = 0 \\ exp\left\{-\left[1 + \xi\frac{z-\mu}{\sigma}\right]^{-\xi^{-1}}\right\}, & \xi \neq 0, \ 1 + \xi\frac{x-\mu}{\sigma} > 0 \end{cases} \qquad (1)$$

where μ, σ and ξ are the location, scale, and shape parameters, respectively.

Particular cases of Eq. (1) with $\xi \to 0, \xi > 0, and \ \xi < 0$ correspond to the Gumbel,
Frèchet, and the negative Weibull distributions, respectively. Generally, the value of $\xi$ is
greater than zero for precipitation data, although the distribution of Gumbel is sometimes
adequate.

Several methods have been developed for the estimation of the parameters of

GEV distributions. These include the method of moments by Christopeit (1994), the less
influenced method of L-moments (Hosking, 1990; Hosking and Wallis, 1997); the
Bayesian method by Smith and Naylor (1987), Coles and Tawn (2005). The most popular
method is the maximum likelihood method (Smith and Naylor, 1987; Unkašerić and
Tošić, 2009), which has the advantage of allowing the addition of fitting co-variables
(such as trends, cycles or physical variables) (Katz et al., 2002). The detailed procedure
of these methods summarized by the El Adlouni et al. (2007), Kioutsioukis et al. (2010),
and Kharin et al. (2013). In this study, the maximum likelihood method is used to
estimate the parameters of the GEV distribution. The regression parameters of
$\mu(t), \sigma(t), and \ \xi(t)$ are the location, scale, and shape respectively. The parameters of the
likelihood function, given n observations $\{(t_1, z_1), (t_2, z_2), ......, (t_n, z_n)\}$ at period $t_i$ at which
the greatest $z_i$ is acquired, is provided by

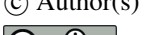



$L\left(\theta|t_t, z_t\right) = \prod_{i=1}^{m} g[z_i;\ \mu(t_i), \sigma(t_i), \xi(t_i)]$     (2)
where
$g(z;\ \mu, \sigma, \xi) = \frac{1}{\sigma}\left\{\left[1 + \xi\left(\frac{z-\mu}{\sigma}\right)\right]^{-(1+1/\xi)}\right\}\ exp\left\{-\left[1 + \xi\left(\frac{z-\mu}{\sigma}\right)\right]^{-1/\xi}\right\}$     (3)
The log-likelihood function is
$l\left(\theta|t_t, z_t\right) = -\sum_{i=1}^{m}\left\{log\sigma(t_i) + \left(1 + \frac{1}{\xi(t_i)}\right)\ log\left[1 + \xi(t_i)\left(\frac{z_i - \mu(t_i)}{\sigma(t_i)}\right)\right] + [1 + \right.$
$\left. \xi(t_i)\left(\frac{z_i - \mu(t_i)}{\sigma(t_i)}\right)\right]^{-\frac{1}{\xi(t_i)}}\right\}$     (4)
$\sigma(t_i) > 0$ and $\{1 + \xi(t_i)(z_i - \mu(t_i))/\sigma(t_i) > 0\}$ for i=1, ...., n. For every value of $\xi(t_i)$
that equals to zero, it is important to utilize the suitable limiting form, replacing the GEV
by the Gumbel ($Eq.(1)\ for\ \xi = 0$) log-likelihood function,
$l\left(\theta|t_j, z_j\right) = -log\sigma(t_i) - \frac{z_j - \mu(t_j)}{\sigma(t_j)} - exp\left[-\frac{z_j - \mu(t_j)}{\sigma(t_j)}\right]$     (5)
The maximum likelihood estimate of θ yields the maximization of Eq. (4) and/or Eq. (5).
Rao (1973) estimated the confidence intervals for the selected return periods using the
delta method. Figure 1 shows the regression, model fits and estimated the return values of
monthly maximum temperatures.

We implement this GEV analysis to study the minimum and maximum surface

temperatures and precipitation as simulated by CMIP5 models in the historical
experiments (years 1901-2005), CRU observations, and experiments for the 21st century
(years 2006-2100) with four different radiative forcing scenarios.
**3. Results**
**3.1 CMIP Historical and CRU temperature extremes**





The spatial distribution of extremes for the Historical runs in India during 1901-
2005 is presented by showing maximum and minimum temperature extremes with
different return time periods are shown in Figure 2. The top and bottom panels show
maximum and minimum extremes respectively with return periods of 10, 20 and 50 years,
denoted as $T_{(max,10)}$, $T_{(max,20)}$, and $T_{(max,50)}$ for maximum temperatures and $T_{(min,10)}$, $T_{(min,20)}$,
$T_{(min,50)}$, for minimum temperatures respectively. The regional mean value for each return
time period is mentioned at the top of each plot. The mean values indicate high warm
extreme temperature conditions in India with average values of 34.89, 36.42, and 38.14$^{o}$C
for $T_{(max,10)}$, $T_{(max,20)}$, and $T_{(max,50)}$ respectively. The mean CRU extreme regional values
are 34.80, 36.46, and 38.42$^{o}$C for the 10, 20, and 50 year periods (Figure not shown).
$T_{(max,10)}$ and $T_{(max,20)}$ show the most evident warm extremes over Northwest and North-
central regions. These extreme regions extend to the Interior peninsula at $T_{(max,50)}$. Similar
extreme warm surface temperatures are observed over the northwestern part of India
(Gadgil, 2018). These three regions show maximum extremes with return values all
above 40$^{o}$C, while the Western Himalaya region exhibits the lowest maximum
temperature extremes at about 10$^{o}$C. At $T_{(max,10)}$ large cold extremes cover most parts of
the Western Himalaya region and slowly turn to warming extremes at $T_{(max,50)}$. The
minimum temperature extremes show large variations over India except for the Western
Himalaya region. The mean value of minimum temperature extreme over the entire
region in India is 7.75, 4.19, and -1.57$^{o}$C for three (10, 20 and 50-year) return periods,
respectively. More extreme cold changes are observed in Figure 2 over the northeastern
and western regions of India, and cold temperature extremes drop from 7$^{o}$C to -20$^{o}$C for




10 and 50 years period. The warmer and colder extremes of the minimum temperature are
observed over southern and northern parts of India respectively.

**3.2 CMIP Historical and CRU changes in temperature extremes**

The spatial differences between CMIP and CRU warm and cold temperature

extremes for the three return estimates of 10, 20, and 50 year periods are shown in Figure
3. The upper and lower panels display the changes in warm and cold temperature
extremes for three time periods respectively. The positive (red color) and negative (blue
color) values in these diagrams indicate the warmest and coldest Historical extremes for
the three different periods.

The difference between the warm extremes decreases slightly from the 10 to 50-

year period over central and northern parts of India. Warm and cold bands are clearly
observed over the southern regions of the warm extreme difference map. Looking at the
cold extreme differences, a cold band (with a magnitude of ~4.5$^{\circ}$C) is observed in the
northwest region of India for the 50-year period, indicating that the CRU cold extremes
are warmer than those of CMIP5 historical runs. The regional mean value decreases from
0.14 to -0.20$^{\circ}$C for warm extremes and decreases from -0.55 to -0.95$^{\circ}$C for cold extremes
from 10 to 50 year periods. From Figure 3, the magnitude of the difference of cold
extremes is little larger than those of the warm extremes for all three return periods over
India. The mean value of warm and cold extreme differences are less than a degree
indicating a fairly good agreement between the Historical and CRU temperatures for the
three different return periods. Kharin et al. (2005, 2007) observed that the temperature
differences between CMIP5 multi-model and ERA-Interim are generally larger for cold
extremes than for warm extremes during the period from 1986 to 2005. Table 2




summarizes the warm and cold extreme temperature mean values for the 10, 20, and 50-
year periods of each region for the CRU, Historical, as well as the differences between
the two. It is evident from the table that the maximum warm extreme mean temperature is
observed in the Interior Peninsula over the Historical ensemble and CRU temperatures
for the 20- and 50-year return periods.
**3.4 Future climate extreme changes in CMIP5 projections**

The spatial GEV distribution for three different return values of 10, 20, and 50

years estimated from CMIP5 maximum temperatures of different RCP scenarios (RCP
2.6, 4.5, 6.0, and 8.5) for the period 2006-2099 are shown in Figure 4. All RCPs suggests
comparable spatial distributions of maximum temperatures over the three different
periods.  The spatial distributions of warm extremes for all RCPs look similar in the 50-
year period. Moderately warm regional mean temperature changes are observed in
RCP2.6 and RCP8.5 scenarios at about 1.15, 1.28, and 1.28$^{\text{o}}$C for the three (10, 20 and 50
year) periods, respectively. In RCP2.6, the warm temperature extremes are observed in
northwest (NW) and north central (NC) regions in the 10-year period, while warm
extremes cover three regions (NW, NC, and IP) in the 20-year period, and most of the
regions in India in the 50 year period. In RCP8.5 the maximum temperatures are
observed in most of the Indian regions with regional means of 39.96, 39.99, and 41.18$^{\text{o}}$C
for the three (10, 20, and 50-year) return periods, respectively. Maximum extreme
temperatures of about ~44$^{\text{o}}$C are observed in several grids throughout India under (RCP
2.6 and 8.5) CMIP5 experiments in the 20 and 50 year return periods. Similar extreme
temperatures reach values of around 46$^{\text{o}}$C in large areas of northwest and Interior
peninsula regions over equatorward of 25$^{\text{o}}$. All simulations demonstrate an ascent of



more than ~3.5$^{o}$C over three regions (NW, NC, and IP), and a warming of more than 2$^{o}$C
over the western Himalayan region in the 50 year period.

The spatial distribution of cold temperature extremes during the 21$^{st}$ century

under the RCP scenarios (RCP 2.6, 4.5, 6.0, and 8.5) for the three different time periods
over India are shown in Figure 5. The regional mean values of cold extremes have
consistently decreasing trends in all RCP scenarios. The northwest, western Himalayas,
and northeast are the main regions exhibiting diminishing trends in all three return
periods. The mean value of cold extremes for the 50-year period is ~7$^{o}$C higher than the
20-year period for RCP2.6. For the other concentration pathways (RCP 4.5, 6.0 and 8.5),
the projected increase in cold temperature extremes ranges from 2.5$^{o}$C to 2.8$^{o}$C, and 3.3$^{o}$
C to 3.9$^{o}$C over the period 10 to 20 and 20 to 50-year return periods, respectively. Note
that the positive changes of about ~5$^{o}$ C in temperature are observed in the RCP8.5
experiment in 21st century relative to the 1901-1960 historic period (Basha et al., 2017).
The cold temperature extreme slowly decreases with latitude from south to north of India
in all RCP scenarios. The magnitude at the southern tip of India is about 20$^{o}$C, which
decreases to -23$^{o}$C over the northern tip. The maximum regional cold extreme value at
about 12.73$^{o}$C is observed in RCP8.5 for a 10-year period, while the minimum at about -
0.99$^{o}$C is observed in RCP2.6 for 50-year return period.
**3.5 Temperature extremes inter-model uncertainty in CMIP5 projections:**

The variability of the warm and cold temperature extremes over India can be

shown by standard deviations as shown in Figures 6 and 7, which depict the spatial
distributions of standard deviations for three different time periods (10, 20, and 50-year)
of warm (T$_{max}$) and cold (T$_{min}$) extremes projected in the four different scenarios (RCP2.6,





4.5, 6.0, and 8.5), respectively. The spatial map in Figure 6 indicates the maximum to be
in the southern part of Interior Peninsula (IP), while the second maximum (relatively
weak) is at the Western Himalaya (WH) region in RCP2.6 at the 50-year period. The
standard deviation of warm extremes is larger in the 50-year period compared to the 10-
and 20-year periods especially in the southern part of India in all RCP scenarios. The
maximum mean value is about $0.75^{\circ}$C in RCP8.5 (10-year period), whereas the minimum
value is observed in RCP2.6 (50-year return value) at about $0.33^{\circ}$C. The standard
deviations change in small increments across different scenarios for all return periods.
For example, the standard deviation changes in 20-year return values are 0.47, 0.45, 0.41,
$0.49^{\circ}$C under RCP2.6, 4.5, 6.0, and 8.5 scenarios, respectively.

The spatial distribution of different CMIP5 experiments for three different time

periods (10, 20, and 50-year) return values of cold extreme ($T_{min}$) standard deviations are
shown in Figure 7. A distinct feature of warm bias (up to $3.5^{\circ}$ C) in eastern and western
regions of India is observed in all scenarios at 20- and 50-year periods. In cold extremes,
the 50-year return period standard deviation is higher compared to other return values
under RCP2.6. The maximum mean value of $T_{min,50}$ is about $2.29^{\circ}$C in RCP2.6, while the
minimum value ($T_{min,10}$) is observed in RCP8.5.  The cold extremes have a larger
variability comparing to warm temperature extremes. The mean maximum value of warm
temperatures ($T_{max,50}$) is almost three times as large as the $T_{min,50}$ in RCP2.6. The
variability of warm extremes (given by the standard deviation) are spatially fairly
uniform in all the return periods, which is not the case for cold extremes under CMIP5
scenarios. Recent observational (Lee et al., 2014) and modeling (Kharin et al., 2007,
2013) studies have reported larger variability of warming in cold extremes compared to





warm extremes across different return periods. This indicates that variability in cold

temperature extremes is larger than those of warm temperature extremes over India.

**4. Precipitation extremes**

**4.1 Historical and CRU precipitation extremes and differences**

The spatial variations of Historical (top panel), CRU (middle panel), and the

differences between the two (bottom panel) of extreme precipitation for three different

return periods (10, 20, and 50-year) are shown in Figure 8. The three (10, 20, and 50-

year) periods of precipitation extremes are computed from the GEV procedure by using

monthly precipitation grids. From Figure 8, precipitation extremes increase significantly

from the 10 to the 50-year period in both Historical and CRU observations. In

CMIP5_historical runs the extreme precipitation appears to have a positive trend in the

Interior Peninsula, which extends slightly into North Central (NC) part of India. The

maximum trends, however is concentrated in the IP region. In the case of CRU, the

increasing trend is observed over the IP and NC regions for the 20-year period, which

also extends to most parts of India except for the southern tip and the Western Himalayan

regions for the 50-year period. A widespread increase in extreme precipitation is

observed in CRU for the 50-year period over the IP, NC, WC and EC regions. The

differences between Historical and CRU extreme precipitations indicate that the CRU

extreme values are slightly higher over the IP and NC, while Historical is slightly higher

in the northern and southern parts of India for the 10- and 20-year periods. In the 50-year

period, precipitation is higher in the Historical runs compared to CRU over the Interior

Peninsula, Western Himalayan regions. However, extreme precipitation is lower in the

Historical runs, in the northwest and extending to northwest and extending to north-





296 central regions of India. The regional mean differences are -11.89%, -11.33% and 4.69%

297 for all three (10, 20, and 50-year) periods, respectively.

298  The multi-model extreme precipitation differences for the 10-, 20-, and 50-year

299 return periods during the period 2006-2100 for each CMIP5 scenarios (RCP2.6, 4.5, 6.0,

300 and 8.5) relative to the 1901-2005 historical periods are shown in Figure 9. The

301 northwestern region has the greatest decrease in all CMIP5 scenarios for all three return

302 periods, which implies that the warmest region has the greatest decrease in extreme

303 precipitation in future projections. The maximum mean difference is about ~23% in

304 RCP8.5 for the 50-year return period. In comparison, future projections of extreme

305 precipitation are slightly higher than Historical ones in the northern and some regions

306 within Interior Peninsula. However, the Historical precipitation extremes are dominant in

307 the 50-year period, and to a smaller extent in the 10-year period. The regional mean

308 changes of extreme precipitation for the 50-year period are -10.4%, -12.9%, -4.3%, and -

309 22.9% under the RCP2.6, 4.5, 6.0, and 8.5 scenarios, respectively. From Figure 9, the

310 regional mean changes of future precipitation extremes are 1.9% and 5.9% in RCP2.6

311 (20-year period) and RCP6.0 (20-year period), respectively. Shashikanth et al. 2017 also

312 found that significant changes in monsoon precipitation extremes during a 30-year period

313 (2081-2100) compared to the historic period.

314 **5. Conclusions**

315  We have assessed the Historical and CRU precipitation and temperature extremes

316 and likely future changes within them throughout India. We quantified the warm and cold

317 temperatures as well as precipitation extremes of CMIP5 for all Representative

318 Concentration Pathway scenarios (RCP2.6, 4.5, 6.0, and 8.5) for the future using a



statistical model of climate extremes based on GEV distributions for the three return
periods (10, 20, and 50-year). The most important findings of our analysis are
summarized as follows:

Extreme warm values in Historical $T_{max}$ in India appear to be rather moderate.

The regional means of extreme maximum temperatures are 34.89, 36.42, and 38.14 $^{o}$C for
all three (10, 20, and 50-year) return periods, respectively, while the minimum extreme
temperatures are 7.75, 4.19, -1.47 $^{o}$C for those same return periods. Comparing the 10- to
50-year return periods, the warm extremes increase at about ~3 $^{o}$C over northwestern,
north central, and Interior peninsula regions. Cold extremes are decreased ~5 $^{o}$C
especially over the eastern and western regions of India.

The regional relative mean differences of Historical and CRU $T_{max}$ extremes are

0.14, 0.01 and -0.20 $^{o}$C for the three (10, 20, and 50-year) periods, respectively.
Comparing the 10- and 50-year return periods shown that the relative changes of extreme
temperatures decrease in Northwest, North central, and northern part of Interior peninsula,
and increase over lower part of the west coast. The relative mean differences of CRU
cold extremes are slightly higher than those of the Historical runs. The relative mean
differences of cold extremes are -0.55, -0.64, and 0.28 $^{o}$C for the three (10, 20, and 50-
year) periods, respectively. CRU shows more changes in the cold extremes as opposed to
warm extremes compared to the Historical extremes. Regionally, northwestern and
northeastern regions of India show the highest changes.

Future $T_{max}$ extreme temperatures increase in all RCP scenarios compared to

historical temperatures, especially for the 20 and 50 year periods. The regional extreme
mean values increase moderately compared to the historical values at about 1.85 and 2.92



°C in the 50-year period under RCP6.0 and 8.5 scenarios. In the case of $T_{min}$ extreme
mean temperatures of RCP2.6 decrease by nearly 5 °C compared to the historical values,
while the minimum extreme temperature mean in RCP8.5 increase by nearly 4 °C
compared to historical temperatures in 50-year return period. It must be noted that the
effect of increasing radiative forcing under higher concentration pathways is larger on
cold temperatures compared to warm temperatures.

The spatial variability of CRU extreme precipitation rates is substantially larger

compared to Historical extremes in all three return periods. Upon comparing 10-, and 50-
year periods, changes in precipitation extremes are observed in both the location and
scale of the distribution, especially over North Central and Interior Peninsula regions of
India. In the other regions, CRU precipitation extreme changes increase slightly in the
50-year period. The regional mean relative difference of Historical and CRU precipitation
extremes is observed the 50-year period at about -14.6%. It indicates that Historical
precipitation extremes show smaller values compared to CRU in several regions in India.
The past and future differences of extreme precipitation are significantly larger when
comparing to Historical to RCP8.5, implying that increasing radiative forcing under
higher greenhouse gas concentrations may lead to larger changes in precipitation
extremes.
**Acknowledgements**

We acknowledge the GCM modeling groups, the Program for Climate Model

Diagnosis and Inter-comparison (PCMDI), and the WCRP's Working Group on Coupled
Modeling for their roles in making available WCRP CMIP5 multi-model datasets. The

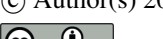



authors would like to thank the National Center for Atmospheric Research (NCAR) for
providing the CRU data.
**Figure captions**
Figure 1. Sample plot of Generalized Extreme Value (GEV) distribution return values,

empirical and modeled fits with 95% confidence level, together with the map of

India divided in the seven regions used in this study.

Figure 2. The historical maximum temperature ($T_{max}$; top panel), and minimum

temperature ($T_{min}$; bottom panel) extremes for 10-year (left), 20-year (middle),

and 50-year (right) periods during 1901-2005.

Figure 3. The difference between CMIP5_historical and CRU maximum temperature

($T_{max}$; top panel), and minimum temperature ($T_{min}$; bottom panel) extremes for

(left) 10-year, (right), 20-year, and (right) 50-year periods during 1901-2005.

Figure 4. The (left) 10-year, (middle) 20-year, and (right) 50-year return values of CMIP5

multi-model mean of warm temperature extremes for the period 2006-2100

under RCP2.6 (1$^{st}$ row), RCP4.5 (2$^{nd}$ row), RCP6.0 (3$^{rd}$ row), and RCP8.5

(bottom row) scenarios, together with the regional average stated on top of each

panel.

Figure 5. The (left) 10-year, (right) 20-year, and (right) 50-year return values of CMIP5

multi-model minimum temperature extremes projected in 2006-2100 under

RCP2.6 (1$^{st}$ row), RCP4.5 (2$^{nd}$ row), RCP6.0 (3$^{rd}$ row), and RCP8.5 (bottom

row) experiments, together with the regional means stated on top of each panel.

Figure 6. The CMIP5 inter-model standard deviations for the 10-year (left), 20-year

(middle), and 50-year (right) return values of warm temperature extremes

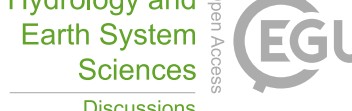



simulated in the RCP2.6 (1$^{st}$ row), RCP4.5 (2$^{nd}$ row), RCP6.0 (3$^{rd}$ row), and

RCP8.5 (bottom row) experiments, respectively.

Figure 7. The CMIP5 inter-model standard deviations for the 10-year (left), 20-year

(middle), and 50-year (right) return values of cold temperature extremes

simulated in the RCP2.6 (1$^{st}$ row), RCP4.5 (2$^{nd}$ row), RCP6.0 (3$^{rd}$ row), and

RCP8.5 (bottom row) experiments, respectively.

Figure 8. The 10-year (left), 20-year (middle), and 50-year (right) return values of

Historical (1$^{st}$ row), CRU (2$^{nd}$ row), and the relative change between Historical

and CRU (%, bottom row) of precipitation extremes during 1901-2005.

Figure 9. The CMIP5 multi-model mean relative change (%) for the 10-year (left), 20-

397        year (middle), and 50-year (right) return values of precipitation extremes

between the historic values in 1901-2005 and the simulated values in 2006-2100

under RCP2.6 (1$^{st}$ row), RCP4.5 (2$^{nd}$ row), RCP6.0 (3$^{rd}$ row), and RCP8.5

(bottom row) scenarios, together with their regional means of relative changes

on top of each panel.











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





Table 1: Historical and CMIP5 different scenarios (RCP2.6, RCP4.5, RCP6.0, and RCP8.5) precipitation and maximum and minimum temperature

| Model Name | Historical 1901-2005 | | CMIP5 2006-2099 | | | | | | | |
| | | | Stem | | | | Pr | | | |
| | Stem | Pr | RCP2.6 | RCP4.5 | RCP6.0 | RCP8.5 | RCP2.6 | RCP4.5 | RCP6.0 | RCP8.5 |
|---|---|---|---|---|---|---|---|---|---|---|
| CCSM4 | Y | Y | Y | Y | Y | Y | Y | Y | Y | Y |
| CNRM-CM5 | Y | Y | Y | Y | Y | Y | Y | Y | Y | Y |
| CSIRO-MK3 | Y | Y | Y | Y | Y | Y | Y | Y | Y | Y |
| CanESM2 | Y | Y | Y | Y | Y | Y | Y | Y | Y | Y |
| GFDL-CM3 | Y | Y | Y | Y | Y | Y | Y | Y | Y | Y |
| GISS-E2-H | Y | Y | Y | Y | Y | Y | Y | Y | Y | Y |
| GISS-E2-R | Y | Y | Y | Y | Y | Y | Y | Y | Y | Y |
| HadGEM2-CC | Y | Y | Y | Y | Y | Y | Y | Y | Y | Y |
| HadGEM2-ES | Y | Y | Y | Y | Y | Y | Y | Y | Y | Y |
| IPSL-CM5A-LR | Y | Y | Y | Y | Y | Y | Y | Y | Y | Y |
| MIROC-ESM | Y | Y | Y | Y | Y | Y | Y | Y | Y | Y |
| MIROC5 | Y | Y | Y | Y | Y | Y | Y | Y | Y | Y |
| MPI-ESM-LR | Y | Y | Y | Y | Y | Y | Y | Y | Y | Y |
| MRI-CGCM3 | Y | Y | Y | Y | Y | Y | Y | Y | Y | Y |
| NorESM1-M | Y | Y | Y | Y | Y | Y | Y | Y | Y | Y |
| BCC-CSM1-1 | Y | Y | Y | Y | Y | Y | Y | Y | Y | Y |
| INMCM4 | Y | Y | Y | Y | Y | Y | Y | Y | Y | Y |
| GFDL-ESM2M | N | N | N | N | N | N | N | N | N | N |
| BNU-ESM | N | N | N | N | N | N | N | N | N | N |
| IPSL-CM5A-MR | N | N | N | N | N | N | N | N | N | N |
| CANESM2 | Y | Y | Y | Y | Y | Y | Y | Y | Y | Y |
| FGOALS-g2 | Y | Y | Y | Y | Y | Y | Y | Y | Y | Y |
| CESM1-CAM5 | Y | Y | Y | Y | Y | Y | Y | Y | Y | Y |






Table 2: CRU and differences between CRU and historical maximum and minimum temperature and standard deviation for seven homogeneous regions for 10-, 20-, and 50-year return periods.

| Regions | CRU: $T_{max}$ | | | CRU: $T_{min}$ | | | CMIP - CRU : $T_{max}$ | | | CMIP - CRU : $T_{min}$ | | |
|---|---|---|---|---|---|---|---|---|---|---|---|---|
| | Avg ± std | | | Avg ± std | | | Avg ± std | | | Avg ± std | | |
| | 10 year | 20 year | 30 year | 10 year | 20 year | 30 year | 10 year | 20 year | 30 year | 10 year | 20 year | 30 year |
| India | 34.80 ± 5.87 | 36.46 ± 6.15 | 38.42 ± 6.83 | 9.77 ± 8.07 | 6.51 ± 8.39 | 1.86 ± 10.04 | 0.14 ± 1.19 | 0.01 ± 1.05 | -0.20 ± 1.46 | -0.55 ± 0.82 | -0.64 ± 0.31 | -0.95 ± 2.12 |
| IP | 37.51 ± 1.59 | 40.12 ± 2.19 | 43.92 ± 3.33 | 15.55 ± 1.92 | 13.72 ± 2.53 | 11.51 ± 3.48 | 0.27 ± 1.06 | 0.09 ± 1.33 | -0.30 ± 2.08 | -0.35 ± 0.29 | -0.75 ± 0.61 | -1.52 ± 1.42 |
| EC | 35.62 ± 1.41 | 36.78 ± 1.63 | 38.08 ± 2.07 | 17.48 ± 2.90 | 15.17 ± 3.37 | 11.67 ± 6.42 | -0.29 ± 2.03 | -0.17 ± 1.96 | 0.01 ± 0.83 | 0.31 ± 0.49 | 0.31 ± 0.49 | 0.61 ± 0.95 |
| NC | 37.91 ± 2.82 | 39.81 ± 3.03 | 41.91 ± 3.44 | 9.76 ± 2.67 | 5.76 ± 3.42 | 0.19 ± 5.32 | 1.33 ± 1.51 | 0.77 ± 1.36 | -0.13 ± 1.43 | -0.28 ± 0.45 | 0.03 ± 1.03 | 0.57 ± 1.57 |
| NW | 38.13 ± 4.26 | 39.37 ± 4.33 | 40.47 ± 4.42 | 8.16 ± 3.06 | 3.98 ± 1.72 | -1.54 ± 1.88 | 1.14 ± 0.56 | 0.85 ± 0.53 | 0.49 ± 0.51 | -0.89 ± 0.69 | -1.32 ± 0.76 | -2.75 ± 3.86 |
| WC | 34.59 ± 2.27 | 35.83 ± 2.59 | 37.41 ± 3.18 | 16.44 ± 2.77 | 14.43 ± 3.03 | 11.63 ± 5.03 | -0.33 ± 0.93 | 0.21 ± 1.22 | 1.37 ± 1.57 | -0.01 ± 0.58 | -0.64 ± 0.26 | -2.28 ± 2.09 |
| NE | 30.46 ± 5.48 | 31.44 ± 5.65 | 32.31 ± 5.86 | 6.69 ± 5.78 | 1.10 ± 4.45 | -8.99 ± 6.68 | -1.33 ± 1.96 | -1.42 ± 2.01 | -1.54 ± 2.03 | -0.93 ± 0.96 | -0.69 ± 1.38 | -0.11 ± 2.03 |
| WH | 18.14 ± 5.52 | 19.59 ± 5.36 | 20.74 ± 5.23 | -13.75 ± 7.14 | -15.71 ± 8.39 | -17.53 ± 7.32 | -2.28 ± 1.52 | -1.63 ± 1.15 | -0.89 ± 1.76 | -2.02 ± 0.68 | -1.87 ± 0.68 | -1.67 ± 0.68 |

IP = Interior Peninsula; EC = East Coast; NC = North Central; NW = North West; WC = West Coast; NE = North East; WH = Western Himalayas.






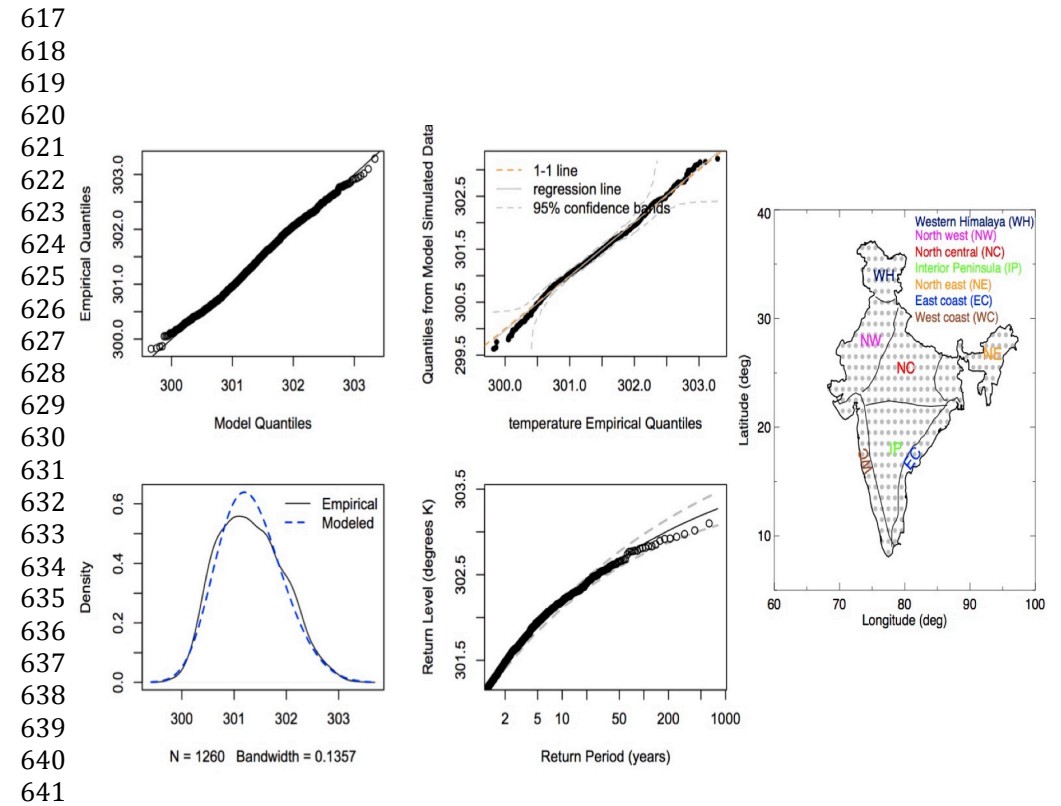

**Figure 1**





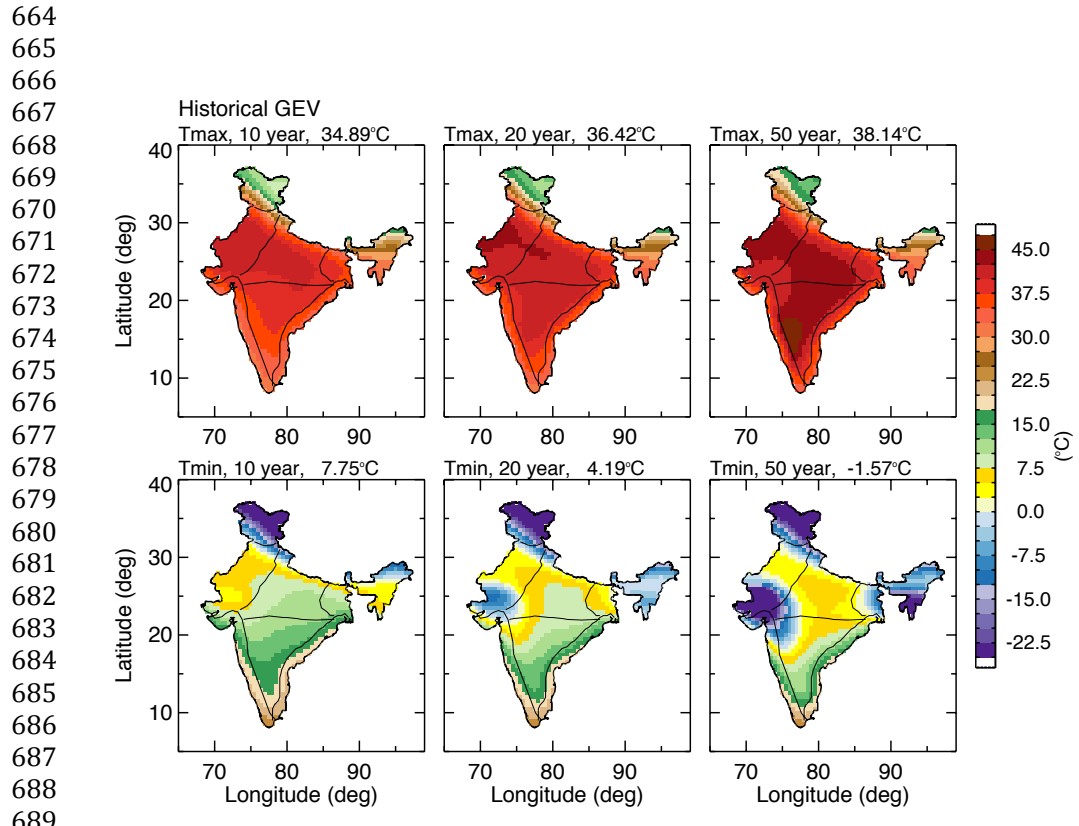

**Figure 2**





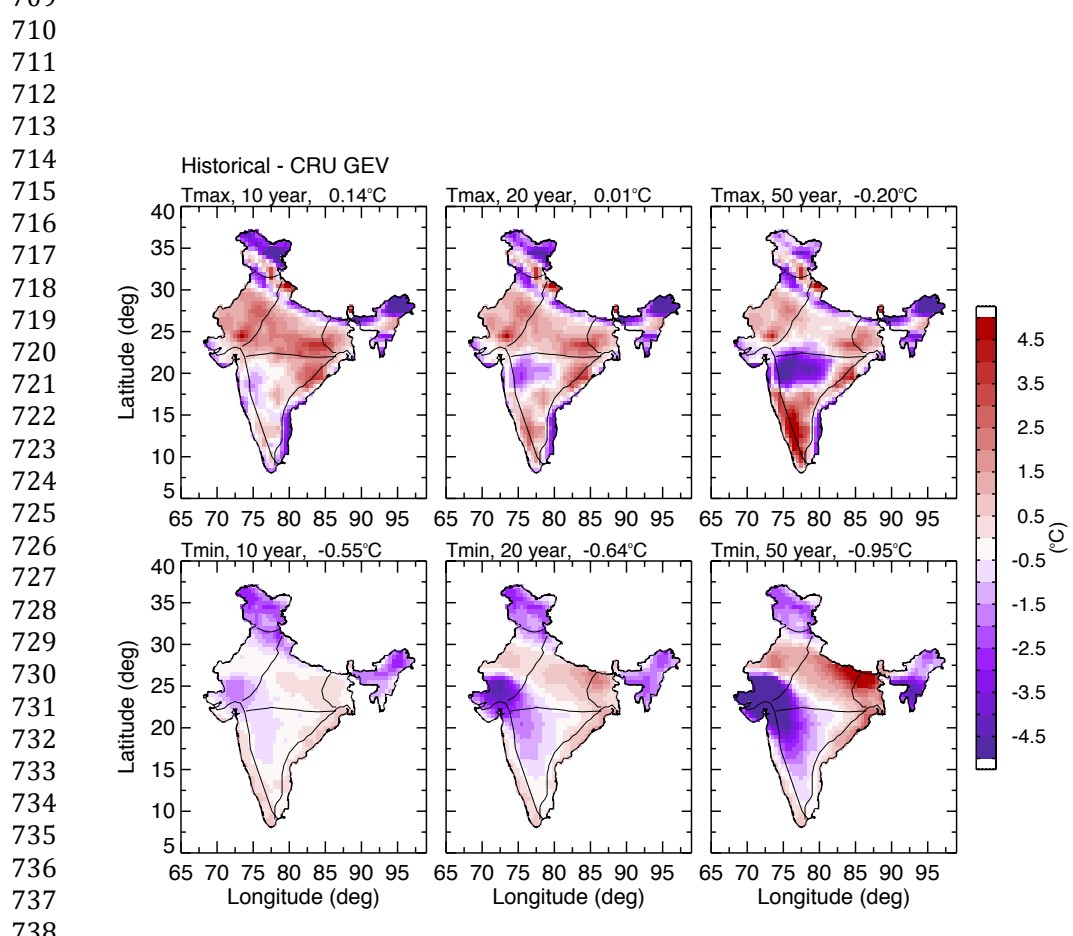

**Figure 3**




**Figure 4**





**Figure 5**





**Figure 6**





**Figure 7**





**Figure 8**




**Figure 9**