# Peer review of "Discussion started: 8 November 2018 © Author(s) 2018. CC BY 4.0 License."

_Hydrology and Earth System Sciences, 2018_

## Referee Comment (RC1) · Anonymous Referee #1 · 11 Dec 2018

**REVIEW OF PANGALURU ET AL., 2018, HESS: ESTIMATING CHANGES OF TEMPERATURES AND PRECIPITATION EXTREMES IN INDIA USING THE GENERALIZED EXTREME VALUE (GEV) DISTRIBUTION**

**GENERAL COMMENTS**

Pangaluru and co-authors quantify extreme temperature (maxima and minima) and precipitation over India using climate models with historical forcing and CRU data. They show biases between the two datasets spatially over India. I may not have a lot of comments, but overall I find that the analysis is shallow and I fail to grasp what the significant contribution of this paper is. There is already a lot of literature comparing historical and future temperature using climate models. For example, the IPCC report, but also many other papers, often on a global scale. Therefore, I strongly doubt whether the manuscript in its current (or even updated) form meets the standards of HESS. Here I provide some key points that need to be addressed:

- None of the figures actually provides information on future **changes** in the extremes over India. All that can be seen is that there are biases between the climate models and CRU. This should be analyzed because that is the analysis that was suggested in the title of the paper and even what is concluded on, for example, L316 and L356-359. It would probably also require carrying out bias-correction for the climate models.
- The CRU data is monthly. Most literature for extremes is on daily or subdaily scale. I am not sure if it is very useful to do extreme value analysis on monthly data. Essentially you probably have 2 or 3 months each year that can have the highest temperature or precipitation. You could also take the month which generally has to highest temperature and do a trend analysis. I would like to see some analysis by the authors that showed that the assumption of a GEV-distribution is useful or use references to explain that this is a useful approach.
- Regional averaging: is this done before or after the application of the GEV model? Explain why.
- Extreme value theory is normally used to estimate extremes with return period beyond the window of training data. With 100+ year of data, you could just get the 10, 20 and 50 year return levels directly from your data, without fitting any GEV model, so what is the point in doing that?
- I did not comment on the captions, because the author have made it too difficult for a reviewer to assess them as the captions and the figures are given separately. Please fix this in a revised manuscript.
- The precipitation analysis is carried out much less rigorously compared to the temperature analysis. I think this should be extended.

**SPECIFIC AND TECHNICAL COMMENTS**

*Are the temperatures monthly average temperatures (day+night), or are they monthly averages of daily maxima/minima or something else?*

L31-32: "The GEV statistical distribution is a time-dependent distribution"

*The GEV distribution in itself is not time-dependent. Only if you make the parameter non-stationary (e.g. Wilcox et al., 2018). Later on in the methods the authors (falsely) suggest a time-dependency of the parameters (L136), but I actually think that they applied GEV non-stationary for different time periods, which is different from applying a time-dependent (i.e. non-stationary) method.*

L31-34: "The GEV statistical distribution is a time-dependent distribution with different time scales of variability bounded by a precipitation, maximum ($T_{max}$), and minimum ($T_{min}$) temperature extremes and also assessed their possibility changes are evaluated and quantified over India is presented."

*There is something wrong in this sentence.*

L37-40: "The regional means of historical warm extreme temperatures are 34.89, 36.42, and 38.14 ∘C for three different (10, 20, and 50-year) periods, respectively; whereas the cold extreme mean temperatures are 7.75, 4.19, and -1.57 ∘C."

*Since there has not been given a definition of a regional mean extreme temperature this information is not very helpful in the abstract. The word period is somewhat misleading here. I suggest to use always (throughout the manuscript) include the word 'return' when talking about return periods. Period could just refer to a certain length of time.*

L46-47: "The CRU precipitation extremes are larger than the historical extreme precipitation in all three (10, 20, and 50-year) return-periods."

*What does this mean? CRU is historical precipitation right?*

L56-57: "India is the third most influenced nation by weather related by disasters, which can largely be attributed to both higher occurrences of extreme temperatures and precipitation"

*Does this statement include or exclude the effect of risen $CO_2$-levels in the atmosphere, or is it because of the fact that people simply live very close (perhaps too close) to rivers for example.*

L58-60: Trenberth (2005) showed that climate change due to increased greenhouse gas emissions leads to changes in extreme event behavior in terms of precipitation and temperature all over the world.

*Really? The title of the paper suggest that it looks at uncertainties in hurricanes. Perhaps other references are more appropriate.*

L64: "Jaruskova and Rencova (2008)"

*Not sure why this reference is singled out. There are probably hundreds of papers using GEV. A reference that should probably not be missed is Papalexiou and Koutsoyiannis (2013), since that does global analysis and thus includes India as well.*

L124: "Generally, the value of $\xi$ is greater than zero for precipitation data"

*Can you provide a reference?*

L161: "return time periods"

*I think this should be return period or return time and not return time period*

L215-217: "Moderately warm regional mean temperature changes are observed in RCP2.6 and RCP8.5 scenarios at about 1.15, 1.28, and 1.28 °C for the three (10, 20 and 50 year) periods, respectively."

*Where can I see these numbers back?*

L228: "warming of more than 2 °C over the western Himalayan region in the 50 year period"

*The word 'Warming' suggests a trend in time of 2 degrees Celsius over 50 years, but I think the authors are just discussing the biases between CRU and CMIP5 for the 50-year return period or something like that. Please provide more details of what you actually did and better explain what you mean exactly.*

L322: "Extreme warm values in Historical Tmax in India appear to be rather moderate"

*I could be me just being from a much colder country, but a regional average monthly value of temperature of 38 degrees Celsius is not something I would classify as moderate.*

*Table 2: clearly provide the considered time period in the caption.*

*Figure 1: axes lack units…*

*Figure 2: The word historical is misleading. CRU is historical data, and climate models can simulate the climate of the past, but that does not make them actual history.*

**TECHNICAL CORRECTIONS**

*Notation of italic and non-italic symbols is a mess and not up to the standards as outlined in the HESS manuscript preparation guidelines.*

L60: **'The'** *Generalized Extreme … (this same error occurs at other places as well)*

*Table 2: 30 year should probably be 50 year*

*L331: shown → showed*

**REFERENCES**

Papalexiou, S. M. and Koutsoyiannis, D.: Battle of extreme value distributions : A global survey on extreme daily rainfall, Water Resour. Res., 49(1), 187–201, doi:10.1029/2012WR012557, 2013.

Wilcox, C., Vischel, T., Panthou, G., Bodian, A., Blanchet, J., Descroix, L., Quantin, G., Cassé, C., Tanimoun, B. and Kone, S.: Trends in hydrological extremes in the Senegal and Niger Rivers, J. Hydrol., 566(September), 531–545, doi:10.1016/j.jhydrol.2018.07.063, 2018.

---

## Referee Comment (RC2) · Anonymous Referee #2 · 13 Jan 2019

Review of "Estimating changes of temperatures and precipitation extremes in India using the Generalized Extreme Value (GEV) distribution" by Pangaluru et al.

Pangaluru et al. analyze the spatial distribution of extreme precipitation/temperature return levels over India. To estimate the return levels, they use a GEV distribution. The paper is well written and the content fits HESS. My main concern is that the authors are using a non-stationary GEV model, but they do not even mention non-stationarity. Research has not yet agreed if all parameters of GEV model should be considered non-stationary (Lee et al. 2017). The authors (without convincing the reader) just use non-stationary GEV model and consider that all parameters are non-stationary.

[Figure]

Therefore, I recommend publishing the article once the authors address the followings:

Major concerns: Non-stationarity: It seems that the authors are using a non-stationary GEV distribution while they do not even mention non-stationarity. This is not acceptable. If you are using a non-stationary model, you have to convince the reader that this is the best option. Why are you using a non-stationary model? There is no literature review about Non-stationarity! Why did you consider all three parameters to be non-stationary? Studies have concluded that it is very hard to estimate a non-stationary shape parameter in GEV distribution (see Lee et al., 2017 and references cited there). It's not clear how the authors have dealt with this problem. Literature review: The literature review does not seem to be covering the entire content. More literature review on precipitation extremes is needed. Also, a significant portion of the introduction covers extremes in China while the study area is India. I suggest adding literature review of other regions such as U.S. as well. A recent paper on the extremes of the U.S. is Zarekarizi et al., (2018). Model parameters: If the parameters are non-stationary, the readers need to see the variations in the parameters. I would like to see the parameters (all of them) in all datasets! (Historical, CRU, and all RCPs). Did you estimate the parameters for every dataset (Historical, CRU, and every RCPs)? Did you estimate the parameters once in the historical period and used it for the future? Please explain and show the estimate parameter maps in the revisions. Model choice: Authors need to prove that non-stationary GEV is the best-fit model. I am not convinced why the authors have used non-stationary GEV. It's not clear why the authors chose GEV for extreme precipitation. You can add literature review if other studies have use GEV for extreme precipitation as well. Authors are only using plots to show the goodness-of-fit. This is not enough. Please use 1-2 quantitative measure too (such as AIC, BIC, DIC, RMSE, NSE, etc.) Downscaling: (Line 112) Did you do any downscaling? If yes, explain. if not, explain the reason in the discussion section! Also, say exactly how many cells you have? For figures, have you used smoothing? If yes, what method? Introduction: The introduction needs more literature review on precipitation extremes. Explain in more detail what are the goals of the study?

[Figure]

Minor concerns: It's not clear what program the authors have used to estimate the parameters. Cite papers that have used GEV to model extreme precipitation. In case, a reader is interested to reproduce you research, where can they get the data? Please provide links. Also, I always encourage open source research. Please indicate if you are planning to share data and code? If yes, where can a reader find the data? Explain how you extracted cold extremes from the dataset. Figure captions are generally too short and need to address more details. Line 32-34: Revise, grammatical issue. Line 39: Use the term "return period" Line 46: What is CRU? Spell out or just say observation. Line 56: Grammatical issue. Line 67: What was their conclusion? Line 81: Spell out GCM. Line 90: This is confusing. Please separate data and method. There is no transition from data to method. Line 94: Is the length of CRU data up to 2005 or you chose this period? Line 97: It is not clear if the authors have done "quality checking procedures" themselves or it's available through data sources? Line 106-107: Explain in more detail. Line 114-116: Not clear. Revise please. Line 117: Add a section or subsection here. Line 129: Please look at Lee et al., (2017) in GRL, too. Line 151: What do you mean by "regression"? Line 151: Explain "delta" method. Line 152: As you know, in GEV model, the realizations should be i.i.d. How can you convince the reader that a month is enough to make sure that data are not dependent? You can cite previous papers that have used monthly maximum temperature and precipitation. Figure 1: Please indicate how many points you have and if you are ignoring any outliers? Line 156: Please add all the fitting information. Figure 1: What data are you using for these plots? Historical or CRU? Explain in the caption. It would be good to see this for other datasets too (if possible). Line 161: Please use the term "return period" Line 165: Please explain this in the caption, too. Line 168: "(Figure not shown)". I think it's important to show this figure (if not limited in the number of figures. In that case, you can add it in the responses to this review) Line 209: Revise the title of sub-section. This section is more of analyzing the spatial distribution, not changes. Line 247: standard deviation of what? Estimated return levels? Line 271: I could not find lee et al. (2014) in the list of references. Please make sure all the cited papers are included in

the references. Line 283: You did not analyze trend. Please revise this term. Line 313: Issue with citing style.

Tables: Table 1: Please highlight the rows to indicate the models that you have used in the study. Also, make it clear in the caption too. Table 1: The quality (of both tables) is low. Explain in the caption what "stem" is.

Figures: I suggest separating the map in figure 1 from the rest of figure. They are representing different ideas. What are the gray dots in the map in figure 1? Figure 1, lower right panel: add legend. Expand the figure 1 caption and add more detail especially about the dataset. Figure 2: Explain in the caption that what the numbers above each panel is? This applies to all figures. It is not clear from the caption that what the figure represents (this should be applied to all captions) I suggest adding 6 more panels below these and represent CRU data. Figure 3: Revise the title of the colorbar to make sure the reader understands that this is difference and not absolute values. Is this difference between return periods? Or absolute values. Make it clear in the figure. Figure 4: You could change the colorbar so that the spatial distribution of the data is clearer.

References: Lee, B.S., Haran, M, Keller, K., 2017. Multidecadal Scale Detection Time for Potentially Increasing Atlantic Storm Surges in a Warming Climate. Geophys. Res. Lett. https://doi.org/10.1002/2017GL074606 Zarekarizi, M., Rana, A., Moradkhani, H., 2018. Precipitation extremes and their relation to climatic indices in the Pacific Northwest USA. Clim. Dyn. https://doi.org/10.1007/s00382-017-3888-2

Please also note the supplement to this comment:
https://www.hydrol-earth-syst-sci-discuss.net/hess-2018-522/hess-2018-522-RC2-supplement.pdf

---

## Author Comment (AC1) · 4 Mar 2019

Reviewer #1 replies

Pangaluru and co-authors quantify extreme temperatures (maxima and minima) and precipitation over India using climate models with historical forcing and CRU data. They show biases between the two datasets spatially over India. I may not have a lot of comments, but overall I find that the analysis is shallow and I fail to grasp what the significant contribution of this paper is. There is already a lot of literature comparing historical and future temperature using climate models. For example, the IPCC report, but also many other papers, often on a global scale. Therefore, I strongly doubt whether

the manuscript in its current (or even updated) form meets the standards of HESS. Here I provide some key points that need to be addressed:

Reply: The authors thank the reviewer for his/her comments/suggestions which increases the quality of manuscript drastically. We will partly agree with the reviewer's comment on a lot of literature on historical and future temperature extremes over the globe but very few articles are available over particular regions like India. In this revision, we have provided point-by-point replies to the reviewer's comments and information concerning how we handled the revised manuscript while also considering the other reviewer comments/suggestions.

None of the figures actually provides information on future changes in the extremes over India. All that can be seen is that there are biases between the climate models and CRU. This should be analyzed because that is the analysis that was suggested in the title of the paper and even what is concluded on, for example, L316 and L356-359. It would probably also require carrying out bias-correction for the climate models.

Reply: The authors apologize for this mistake. We applied the bias correction and downscale method (BCSD) and discussed in the revised manuscript. For our analysis purpose, we consider the common period of Historical model and CRU observational datasets (1901-2005) and adjust the biases in CMIP5 RCPs (2.6, 4.5, 6.0, and 8.5) output for projected the time period (2006-2099) by assuming a constant model bias. We also included these lines are incorporated in the revised manuscript. Our main intension was to show the temperature and precipitation extremes of observational and model datasets during the 20th and 21st centuries using generalized extreme value (GEV) statistical distribution.

The CRU data is monthly. Most literature for extremes is on daily or sub-daily scale. I am not sure if it is very useful to do extreme value analysis on monthly data. Essentially you probably have 2 or 3 months each year that can have the highest temperature or precipitation. You could also take the month which generally has to highest temperature

and do a trend analysis. I would like to see some analysis by the authors that showed that the assumption of GEV-distribution is useful or use references to explain that this is a useful approach.

Reply: Thanks for your comment. We used monthly average daily maximum and minimum temperatures and precipitation datasets and we corrected in the revised manuscript. We have gone through several published papers, from the literature we found daily, monthly and yearly temperature and precipitation datasets are used for the GEV statistical distribution with different periods. Previous literature we found that:

Wen et al. (2015): They utilized for their GEV analysis purpose the monthly average daily maximum and minimum temperature and precipitation datasets during the period from 1901 to 2005 over China. Ashouri et al. (2016): They examined both the annual maximum precipitation events and precipitation peaks above a certain threshold using CPC and MERRA datasets. Fernado et al. (2006): The datasets are considered monthly maximum for each year, exceedances over large thresholds are used for GEV analysis. Kharin et al. (2013): They considered annual extreme of daily maximum and minimum surface air temperatures of CMIP5 models during the years 1850-2005. Kumar et al. (2017): They examined the annual maximum peak flood data that vary over the period 1957-1989 for 115 bridge sites. Naima et al. (2017): For their GEV analysis purpose the annual maximum of daily precipitation in different regions northern Algeria from 1936 to 2009. Shashikanth et al. (2017): They examined Indian summer monsoon (JJAS) each year peaks are utilized for their analysis purpose. None of the studies explain the non-stationary GEV model spatial structures of temperature and precipitation extremes using Historical model and CRU observations (1901-2005), and all scenarios (RCP2.6, 4.5, 6.0, and 8.5) of CMIP5 models (2006-2099), especially over India.

Regional averaging: is this done before or after the application of the GEV model? Explain why?

Reply: Thanks for your comment. The regional temperature and precipitation mean values are estimated for the 10-, 20-, and 50-year return periods of GEV model results. The regional mean values are estimated using all seven regions over India. The regional mean is the same as the spatial average over India, but we used the word 'regional mean' in the text.

Extreme value theory is normally used to estimate extremes with return period beyond the window of training data. With 100+ year of data, you could just get the 10, 20 and 50 year return levels directly from your data, without fitting and GEV model, so what is the point in doing that?

Reply: Thanks for your comment. To the best of our knowledge, for estimating the extremes with return period the window is little more than the required period. However, we checked the literature most of the researchers are extracted the time periods between 10 to 50 years using little more length than the required time period. In our analysis, we used Historical (1901-2005=105 years) for 2oth century purpose, and RCPs (2006-2099=93 years) used for the 21st-century purpose. I hope this length is enough to estimate the temperature and precipitation extremes for three different (10, 20, and 50-year) periods.

I did not comment on the captions, because the author have made it too difficult for reviewer to assess them as the captions and the figures are given separately. Please fix this in a revised manuscript.

Reply: As per reviewer suggestion, we have added the figure caption at the bottom of each figure in the revised manuscript.

The precipitation analysis is carried out much less rigorously compared to the temperature analysis. I think this should be extended.

Reply: We sincerely appreciate the reviewer's suggestion in this regard. In the revised version we extended non-stationary GEV statistical distributions of precipitation

extremes using different scenarios of CMIP5 models (RCP2.6, RCP4.5, RCP6.0, and RCP8.5) and we have added two figures (See Figures 1 and 2) of extreme precipitation results along with the description. The revised manuscript covers the 20th and 21st-century extreme precipitation non-stationary GEV results using Historical model (1901-2005), CRU (1901-2005), and CMIP model (2006-2099).

SPECFIC AND TECHNICAL COMMENTS

Are the temperatures monthly average temperatures (day+night), or are they monthly averages of daily maxima/minima or something else?

Reply: Thanks for your comment. For our analysis purpose, we utilized monthly average daily maximum and minimum temperatures.

L31-32: " The GEV statistical distribution is a time-dependent distribution" The GEV distribution in itself is not time-dependent. Only if you make the parameter non-stationary (e.g. Wilcox et al., 2018). Later on in the methods the authors (falsely) suggest a time-dependency of the parameters (L136), but I actually think that they applied GEV non-stationary for different time periods, which is different from applying a time-dependent (i.e. non-stationary) method.

Reply: We agree with the reviewer's concern and we used the non-stationary GEV statistical distribution. In the revised manuscript, we discussed the stationary and non-stationary GEV distributions and the importance of the non-stationary method. We also included more information on non-stationary GEV model and we have provided references for important statements.

L31-34: "The GEV statistical distribution is a time-dependent distribution with different time scales of variability bounded by a precipitation, maximum (Tmax) and minimum (Tmin) temperature extremes and also assessed their possibility changes are evaluated and quantified over India is presented" There is something wrong in this sentence.

Reply: We are sorry for the mistake. In the revised manuscript, we have avoided these

flaws sentences.

L37-40: "The regional means of historical warm extreme temperatures are 34.89, 36.42, and 38.14 C for three different (10, 20, and 50-year) periods, respectively; whereas the cold extreme mean temperatures are 7.75, 4.19, and -1.59 C". Since there has not been given a definition of a regional mean extreme temperature this information is not very helpful in the abstract. This word period is somewhat misleading here. I suggest to use always (throughout the manuscript) include the word 'return' when talking about return periods. Period could just refer to a certain length of time.

Reply: We sincerely appreciate the reviewer's suggestion in this regard. We will agree the "regional" word is misleading in the abstract. We modified the sentence accordingly in the abstract. The regional mean values are estimated using all seven regions over India. We have explained how we did the regional mean in the revised manuscript. We have included the word "return" throughout the manuscript in the revised manuscript when talking about return periods.

L46-47: "The CRU precipitation extremes are larger than the historical extreme precipitation in all three (10, 20, and 50-year) return-periods. What does this mean? CRU is historical precipitation right?

Reply: Thanks for your comment. Yes, CRU precipitation extremes are larger than historical model ensemble precipitation extremes. Here, the Climate Research Unit (CRU) is the observations and historical is model datasets. The CRU temperature and precipitation datasets are collected from 1901 to 2005 over land areas, based on daily values from rain gauge measurements provided by more than 4,000 weather stations distributed around the world with the horizontal resolution of 0.5x0.5 grids. The CRU and historical precipitation datasets are considered the same period (1901-2005). Nearly 20 historical models are used for our analysis purpose (see Table 1).

L56-57: "India is the third most influenced nation by weather related by disasters, which can largely be attributed to both higher occurrences of extreme temperatures and precipitation" Does this statement include or exclude the effect of risen CO2-levels in the atmosphere, or is it because of the fact that people simply live very close (perhaps very close) to rivers for example.

Reply: Thanks for your suggestion. We included the statement in the revised manuscript.

L58-60: "Trenberth (2005) showed that climate change due to increased greenhouse gas emissions leads to changes in extreme event behavior in terms of precipitation and temperature all over the world. Really? The title of the paper suggest that it looks at uncertainties in hurricanes. perhaps other references are more appropriate.

Reply: We also agree your suggestion, now we added the relevant reference in the revised manuscript.

L64: "Jaruskova and Rencova (2008)" Not sure why this reference singled out. There are probably hundreds of papers using GEV. A reference that should probably not be missed is Papalexiou and Koutsoyiannis (2013), since that does global analysis and this includes India as well.

Reply: We agree with the reviewer's concern in this regard. Papalexiou and Koutsoyiannis used maximum rainfall datasets of 15,137 records from all over the world, with lengths varying from 40 to 163 years. We modified the manuscript and we have provided this reference in the revised manuscript. Thanks for giving the reference, which helped while revising the manuscript.

L124: "Generally, the value of $\xi$ is greater than zero for precipitation data" Can you provide reference?

Reply: Thanks for your suggestion. In the revised manuscript we added the reference.

L161: "return time periods" I think this should be return period or return time and not return time period.

Reply: Thanks for your suggestion. In the revised manuscript we used the term "return period" throughout manuscript.

L215-217: " Moderately warm regional mean temperature changes are observed in RCP2.6 and RCP8.5 scenarios at about 1.15, 1.28, and 1.28oC for the three (10, 20 and 50 year) periods, respectively." Where can I see these numbers back?

Reply: Thanks for your comment. In the revised version of the manuscript, moderately warm spatial mean temperatures are observed in RCP8.5 than RCP2.6 scenarios at about (37.51-36.36=1.15), (39.28-38.00=1.28), (41.18-39.90=1.28oC) for the three (10, 20, and 50-year) return periods, respectively. The spatial mean value over India for each return period is mentioned on the top of each panel in Figure 5 (Figure 4 in the old version).

L228: " warming of more than 2oC over the western Himalayan region in the 50 year period." The word 'Warming' suggests a trend in time of 2 degrees Celsius over 50 years, but I think the authors are just discussing the biases between CRU and CMIP5 for the 50-year return period or something like that. Please provide more details of what you actually did and better explain what you mean exactly.

Reply: Thanks for your comment. We mean to say that compare to the CRU warm extremes, the RCP8.5 simulation fitted warm extreme results are more than 3oC over three regions (NW, NC, and IP), and 1.5-2oC higher over the western Himalayan region in the 50-year return period. The results indicate that rising temperature in the 50-year return period may be due to global warming or other external forcings may influence the long-term precipitation pattern, in addition, an increase in the frequency of drought severity in several regions over India. These are discussed clearly in the revised manuscript.

L322: " Extreme warm values in Historical Tmax in India appear to be rather moderate." I could be just being from a much colder country, but a regional average monthly value of temperature of 38 degrees Celsius is not something I would classify as moderate.

[Figure]

Reply: We agree with the reviewer's concern in this regard. We modified the sentence in the revised manuscript.

Table 2: clearly provide the considered time period in the caption.

Reply: Thanks fro your suggestion. We have added the time period in the Table 2 caption in the revised manuscript.

Figure 1: axes lack units...

Reply: Thanks for your suggestion. We have added the units in Figure 1 in the revised manuscript.

Figure 2: The word historical is misleading. CRU is historical data, and climate models can simulate the climate of the past, but that does not make them actual history.

Reply: We agree with the reviewer's concern in this regard. We changed has historical multi-model ensemble maximum temperature in the revised manuscript.

TECHNICAL CORRECTIONS

Notation of Italic and non-italic symbols is a mess and not up to the standards as outlined in the HESS manuscript preparation guidelines.

Reply: In the revised manuscript we followed the format of HESS journal.

L60: 'The' Generalized Extreme... (this same error occurs at other places as well)

Reply: We are sorry for the mistake and we corrected in the revised manuscript.

Table 2: 30 year should probably be 50 year

Reply: Sorry for the mistake. We corrected the year in Table 2 in the revised manuscript.

L331: shown -> showed Reply: We are sorry for the typo mistake and now corrected in the revised manuscript.

REFERENCES Papalexiou, S. M. and Koutsoyiannis, D.: Battle of extreme value distributions: A global survey on extreme daily rainfall, Water Resour. Res. 49(1), 187-201, doi:10.1029/2012WR012557, 2013.

Wilcox, C., Vischel, T., Panthou, G., Bodian, A., Blanchet, J., Descroix, L., Quantin, G., Casse, C., Tanimoun, B. and Kone, S.: Trends in hydrological extremes in the Senegal and Niger Rivers, J. Hydrol., 566(Septemeber), 531-545, doi:10.106/j.jhydrol.2018.07.063, 2018.

We thank once again for providing thorough review, guidance, and advice throughout the preparation of the paper. Your help and experience were indispensable in pushing us forward to improve our work.

[Figure]

**CMIP5 Precipitation GEV**

Fig. 1. Spatial structures of CMIP5 multi-model ensemble mean of warm extremes for (left) 10-year, (middle) 20-year, (right) 50-year return periods during 2006-2100 under RCP2.6 (1st row), RCP4.5 (2nd row),

CMIP5 Precip GEV STD

**Fig. 2.** The CMIP5 inter-model estimated return levels standard deviations for the 10-year (left), 20-year (middle), and 50-year (right) return values of annual maximum of monthly precipitation extremes simula

---

## Author Comment (AC2) · 4 Mar 2019

Reviewer #2 replies Pangaluru et al. analyze the spatial distribution of extreme precipitation/ temperature return levels over India. To estimate return levels, they use a GEV distribution. The paper is well written and content fits HESS. My main concern is that the authors are using a non-stationary GEV model, but they do not even mention non-stationary. Research has not yet agreed if all parameters of GEV model should be considered non-stationary (Lee et al. 2017). The authors (without convincing the reader) just use non-stationary GEV model and consider that all parameters are non-stationary. Therefore, I recommend publishing the article once the authors address

the following: Reply: We appreciate your thoughtful review and appreciating the actual content of the manuscript by offering constructive suggestions and comments, which made us to improve the manuscript significantly. During revision, we tried our best to provide all the details for your views/suggestions and hope you will be satisfied with the revised manuscript. Details of the revision are given below: We agree with the reviewer's concern about the non-stationary GEV model. In the revised manuscript we discussed the stationary and non-stationary GEV statistical distributions and the importance of the non-stationary GEV model and we provided references for important sentences. Major concerns: Non-stationary: It seems that the authors are using a non-stationary GEV distribution while they do not even mention non-stationary. This is not acceptable. If you are using a non-stationary model, you have to convince the reader that this is the best option. Why are you using a non-stationary model? There is no literature review about non-stationary! Why did you consider all three parameters to be non-stationary? Studies have concluded that it is very hard to estimate a non-stationary shape parameters in GEV distribution (see Lee et al., 2017 and references cited there). It's not clear how the authors have dealt with this problem.

Reply: We sincerely appreciate the reviewer's suggestion in this regard. In the revised manuscript we clearly mentioned the non-stationary method and we added literature regarding the non-stationary GEV statistical distribution. Adoulni et al. (2007) also mentioned non-stationary GEV model is an efficient tool and it takes consideration the dependencies between extreme value random variables or the temporal evolution of the climate. However, the frequency of extremes is likely to change in response to changes in climate (IPCC, 2007). Coles and Dixon (1999) modify the likelihood function by introducing a penalty term to restrict the shape parameter values to the range for which the GEV distribution has finite mean. Kharin and Zwiers (2005) estimated the annual extreme temperature return values from a fitted GEV distribution with time-dependent location and scale parameters. They found that changes in temperature extremes are largely associated with changes in location parameter of the distribution of annual extremes without substantial changes in the shape parameter over most of

the globe. We also discussed the non-stationary GEV distribution method and importance of the non-stationary method in the revised manuscript.

Literature review: The literature review does not seem to be covering the entire content. More literature review on precipitation extremes is needed. Also, a significant portion of the introduction covers extremes in China while the study area is India. I suggest adding literature review of other regions such as U.S as well. A recent paper on the extremes of the U. S. is Zarekarizi et al. (2018).

Reply: We agree with the reviewer's concern in this regard. In the revised manuscript we included more literature on extreme precipitation studies and we reduced the text on extremes over China in the introduction. As per your suggestion, we discussed the extremes over the USA and we added the Zarekarizi et al. (2018) in the revised manuscript. Thanks for providing the recent work on extremes studies.

Model parameters: In the parameters are non-stationary, the readers need to see the variations in the parameters. I would like to see the parameters (all of them) in all datasets! (Historical, CRU, and all RCP's). Did you estimate the parameters for every dataset (Historical, CRU, and every RCPs)? Did you estimate the parameters once in the historical period and used it for the future? Please explain and show the estimate parameter maps in the revisions.

Reply: Thanks for your suggestion. We prepared for Historical, CRU, and all RCPs of CMIP5 datasets of GEV results. Totally, six plots of extreme temperatures and six plots for precipitation and it is not possible to add all in the manuscript. So, we showed one figure in the text and the remaining figures are attached at the bottom of the reviewer replies (see Figures: Pfit1-5).

Model choice: Authors need to prove that non-stationary GEV is the best-fit model. I am not convinced why the authors have used non-stationary GEV.

Reply: Thanks for your comment. Generally and commonly used statistical models

are stationary and non-stationary models. The stationary model, a linear trend in the location and linear trends in both location and scale parameters. For non-stationary models, the estimated return values are calculated at the center of the periods (Grigory et al. 2011). During the recent decades' concern on climate change, some key developments for extending the concepts of the return period with extreme events under non-stationary conditions have appeared in the statistical and climate change (e.g., Wigley 2009; Cooley 2009, 2013). Later Cheng et al. (2014) used stationary and non-stationary GEV model simulations using annual monthly maximum temperatures and they conclude that non-stationary simulations are better than the stationary simulations. So, after going through all the literature and we opted non-stationary GEV model is good for our analysis. I mentioned some of them only, you can find more on this in the Introduction section. All these literature we incorporated and discussed in the revised manuscript.

It's not clear why the authors chose GEV for extreme precipitation. You can add literature review if other studies have use GEV extreme precipitation as well.

Reply: Thanks for your suggestion. In the revised manuscript we have added more literature regarding extreme precipitation in the introduction section.

Authors are only using plots to show the goodness-of-fit. This is not enough. Please use 1-2 quantitative measure too (such as AIC, BIC, DIC, RMSE, NSE, etc.).

Reply: Thanks for your comment. From the literature, we found the best fitting model is the likelihood ratio test and this method is used for our analysis purpose. Simkova and Picek used estimation methods (L-, LQ-, TL-moment, and maximum likelihood) to estimate high quantiles of the Generalized extreme-value distribution (GEV) and Generalized Pareto distribution (GPD) considering various sample size, the shape parameter, and probability. The simulation study revealed that the L-moment and maximum likelihood methods provide the best high quantile estimates of the GPD and GEV distributions. The more description of the likelihood function has given in the revised

manuscript.

Downscaling (Line 112) Did you do any downscaling? If yes, explain. if not, explain the reason in the discussion section! Also, say exactly how many cells you have? For figures, have you used smoothing? If yes, what method?

Reply: Thanks for your comment. The output for both historic and different CMIP5 RCPs outputs are available on different spatial scales, which are spatially downscaled to a common grid 1x1 (longitude-latitude). In the present study we have used bias correction and downscale method (BCSD) and clearly mentioned in the revised manuscript. All figures are generated using Interactive Data Language (IDL). We did not apply any smoothing technique while generating the figures. The total number of cells over India is 574. The cell locations are shown by gray color filled circles in Figure 2 in the revised manuscript.

Introduction: The introduction needs more literature review on precipitation extremes. Explain in more detail what are the goals of the study?

Reply: Thanks for your comment. We provided literature on extreme temperatures and precipitation in the revised manuscript. The details of the study are clearly mentioned in the revised manuscript.

Minor concerns: It's not clear what program the authors have used to estimate the parameters.

Reply: Thanks for your comment. We used the GEV package and it is available in the R programming exTremes (Guilleland and Katz, 2011). This sentence is incorporated in the revised manuscript.

Cite papers that have used GEV to model extreme precipitation.

Reply: Thanks fro your suggestion. We have added all the citations wherever it is necessary in the revised manuscript.

In case, a reader is interested to reproduce you research, where can they get the data? Please provide links. Also, I always encourage open source research. Please indicate if you are planning to share data and code? If yes, where can a reader find the data?

Reply: Thanks for your suggestion. For our analysis purpose we used precipitation, minimum and maximum surface temperatures of CRU (http://www.cru.uea.ac.uk/data), Historical, and different scenarios of CMIP5 models are available at http://cmip-pcmdi.llnl.gov/cmip5/. These datasets are available free for public. Generally, I am providing the code to everybody request via electronic mail.

Explain how you extracted cold extremes from the dataset.

Reply: Thanks for your comment. Generally, cold extremes temperature purpose we utilized the minimum temperatures (Tmin) and we extracted the cold extremes similar to the maximum temperatures. The cold extreme plots are attached at the bottom of the reviewer's replies (see Figures: Pfit1 and Pfit2).

Figure captions are generally too short and need to address more details.

Reply: As per reviewer's suggestion, we provided more information for figure captions in the revised manuscript.

Line 32-34: Revise, grammatical issue.

Reply: We are sorry for the mistake. In the revised manuscript, we have taken utmost care in minimizing the grammatical mistakes.

Line 39: use the term "return period"

Reply: Thanks for your suggestion and we change in the revised manuscript.

Line 46: What is CRU? Spell out or just say observation.

Reply: Thanks for your suggestion, which is implemented in the revised manuscript.

Line 56: Grammatical issue

[Figure]

Reply: This sentence has been modified correctly in the revised manuscript.

Line 67: What was their conclusion?

Reply: We have mentioned their conclusion in the revised manuscript.

Line 81: Spell out GCM

Reply: Thanks for your suggestion, we have given the full name of the GCM in the revised manuscript.

Line 90: This is confusing. Please separate data and method. There is no transition from data to method.

Reply: In the revised manuscript, we have separated data (section 2) and method (sub section 2.1).

Line 94: is the length of CRU data up to 2005 or you choose this period.

Reply: Climate Research Unit (CRU) datasets are available globally over land areas for the period 1901 to 2017 with horizontal resolution of 0.5o x 0.5o on monthly basis. For our analysis purpose we have collected from 1901 to 2005. We have given all details and we have provided the link where we can download the CRU dataset in the revised manuscript.

Line 97: Is it not clear if the authors have done "quality checking procedures" themselves or it's available through data source?

Reply: Thanks for your comment. The CRU precipitation and temperatures datasets quality control checked by the University of East Anglia and they provided for the public. We modified the sentence in the revised manuscript.

Line 106-107: Explain in more detail

Reply: Thanks for your suggestion we have incorporated more details of the ensemble member 'r1i1p1' in the revised manuscript.

Line 114-116: Not clear. Revise please

Reply: We agree with the reviewer comment and we removed those lines because we already discussed about the Table 1 in previous paragraph.

Line 117: Add a section or subsection here

Reply: Thanks for your advice we have added the subsection in the revised manuscript.

Line 129: Please look at Lee et al., (2017) in GRL., too.

Reply: We thank to the reviewer for this suggestion and we added description and reference in the revised manuscript. As suggested by the reviewer, we have discussed of four types of GEVs with his reference in the revised version of the manuscript.

Line 151: what do you mean by "regression"

Reply: Sorry for the mistake, now we modified the sentence in the revised manuscript.

Line 151: Explain "delta" method.

Reply: Thanks for your suggestion we have explained the delta method in the revised manuscript.

Line 152: As you know, in GEV model, the realizations should be i.i.d. How can you convince the reader that a month is enough to make sure that data are dependent? You can cite previous papers that have used monthly maximum temperature and precipitation.

Reply: Thanks for your comment. The probability of values selected by this way converges asymptotically to the generalized extreme value distribution (GEV), under the assumptions that they are independent and identically distributed (iid). From the literature, Fernado et al. (2006) used annual maximum temperature. They used only one point per year for their analysis. They mentioned it is difficult for seasonal extremes. Some of the points are discussed in the revised manuscript.

Figure 1: Please indicate how many points you have and if you are ignoring any outliers?

Reply: Thanks for your comment. For Figure 1, we used the Historical maximum surface temperatures during the period from 1901-2005 at a particular longitude and latitude grid. The total number of points is 1260, in that we observed very few are outliers, but it varies location to location.

Line 156: Please add all the fitting information.

Reply: Thanks for your suggestion. We explained and discussed about the fitting and confidence level procedures in the revised manuscript.

Figure 1: What data are you using for these plots? Historical or CRU? Explain in the caption. It would be good to see this for other datasets too (if possible).

Reply: Thanks for your suggestion. Figure 1 corresponds to the Historical maximum temperature (Tmax, Units = degrees K) during the period from 1901 to 2005 and we mentioned clearly in Figure 1 caption also in the revised manuscript. As per your suggestion, we have generated some more figures and attached at the bottom of the replies document. The plot names are (Figures: Tfit1, Tfit2, Tfit3, Pfit4, and Pfit5).

Line 161: Please use the term "return period"

Reply: Thanks for your suggestion and we corrected in the revised manuscript.

Line 165: Please explain this in the caption, too

Reply: Thanks for your suggestion and we added in the Figure 2 caption in the revised manuscript.

Line 168: "(Figure not shown)". It is important to show this figure (if not limited in the number of figures. In that case, you can add it in the responses to this review).

Reply: Thanks for your comment. As per reviewer's suggestion, we have added the two

more figures and related discussion on precipitation. So, we don't have room to add some more figures. Here are the CRU GEV maximum and minimum extreme results (see Figure 1).

Line 209: Revise the title of sub-section. This section is more of analyzing the spatial distribution, not changes.

Reply: Changes are made as suggested by the reviewer in the revised manuscript. We changed the sub-section title as "Spatial distribution of CMIP5 future climate extremes" in the revised manuscript.

Line 247: standard deviation of what? Estimated return levels?

Reply: Thanks for your comment. The estimated return level standard deviations for CMIP5 for all scenarios (RCP2.6, RCP4.5, RCP6.0, and RCP8.5). These sentences are incorporated in the revised version of the manuscript.

Line 271: I could not find Lee et al. (2014) in the list of references. Please make sure all the cited papers are included in the references.

Reply: Sorry for the mistake. We provided the details of the Lee et al. (2014) in references list. We checked thoroughly all cited references are given in the reference list.

Line 283: You did not analyze trend. Please revise this term.

Reply: Thanks for your suggestion and we revised in the manuscript.

Line 313: Issue with citing style.

Reply: Sorry for the mistake and we corrected in the revised manuscript.

Tables: Table 1: Please highlight the rows to indicate the models that you have used in the study. Also, make it clear in the caption too.

Reply: Thanks for your comment. We keep what we used in this study only in Table 1

and remaining are deleted from Table 1.

Table 1: The quality (of both tables) is low. Explain in the caption what "stem" is.

Reply: We have modified the Table 1 and Table 2 with best resolution. We explained the "stem" and "Pr" in the revised manuscript.

Figures: I suggest the map in figure 1 from the rest of the figure. They are representing different ideas.

Reply: Thanks for your suggestion. We separated the map in figure 1 and renamed as Figure 2 in the revised manuscript.

What are they gray dots in the map in figure 1?

Reply: Actually the gray dots are grid of the data. We separated from the Figure 1 and in the revised manuscript we named as Figure 2.

Figure 1, lower right panel: add legend

Reply: As suggested by the reviewer, we have added the legend for lower right panel and replaced the new Figure 1 in the revised manuscript.

Expand the figure 1 caption and add more detail especially about the dataset.

Reply: Thanks for your suggestion. We added the dataset details for the Figure 1 caption in the revised manuscript.

Figure 2: Explain in the caption that what the numbers above each panel is? This applies to all figures.

Reply: Thanks for your suggestion. We modified the figure caption in the revised manuscript.

Figure 3: Revise the title of the colorbar to make sure the reader understands that this is difference and not absolute values.

Reply: As per reviewer's suggestion, we have modified the title of the colorbar in the revised manuscript.

Is this difference between return periods? or absolute values. Make it clear in the figure.

Reply: Thanks for your comment. The difference between return periods and mentioned clearly in the text and caption of the Figure 4.

Figure 4: You could change the colorbar so that the spatial distribution of the data is clearer.

Reply: Thanks for your suggestion. We redraw the figure 4 with new colorbar (now Figure 5 in the revised manuscript) and changed the numeration of the figure.

References: Lee, B. S., Haran, M. Keller, K., 2017. Multidecadal scale detection time for potentially increasing Atlantic storm surges in a warming climate. Geophys. Res. Lett. https://doi.org/10.1002/2017GL074606. Zarekarizi, M., Rana, A., Moradkhani, H., 2018. Precipitation extremes and their relation to climate indices in the Pacific Northwest USA. Clim. Dyn. https://doi.org/10.1007/s00382-017-3888-2.

Thanks for giving the references, which helped while revising the manuscript.

We have take care of most of the major concerns in the revised manuscript. We once again thank the reviewer for going through the manuscript carefully and offering potential solutions to improve the manuscript further.

Please also note the supplement to this comment:
https://www.hydrol-earth-syst-sci-discuss.net/hess-2018-522/hess-2018-522-AC2-supplement.pdf
* * *
CRU GEV max and min temperature extremes

Tmax, 10 years, 34.80°C    Tmax, 20 years, 36.46°C    Tmax, 50 years, 38.42°C

Tmin, 10 years, 9.77°C    Tmin, 20 years, 6.51°C    Tmin, 50 years, 1.86°C

**Fig. 1.** Spatial variations of CRU maximum temperature (Tmax: top panel), and minimum temperature (Tmin: bottom panel) extremes for 10-year (left), 20-year (middle), and 50-year (right) return periods during 1

[Figure]

**Fig. 2.** Tfit1: The Generalized Extreme Value (GEV) statistical distribution return period values, empirical and modeled fits with 95% confidence level using Historical ensemble simulations of minimum temper

[Figure]

**Fig. 3.** Tfit2: The Generalized Extreme Value (GEV) statistical distribution return period values, empirical and modeled fits with 95% confidence level using CMIP5 RCP8.5 ensemble simulations of minimum temp

[Figure]

**Fig. 4.** Tfit3: The Generalized Extreme Value (GEV) statistical distribution return period values, empirical and modeled fits with 95% confidence level using CRU maximum temperatures (Tmax, units = degrees K

**CMIP5 RCP26 Precipitation GEV results**

Empirical Quantiles (mm)
Model Quantiles (mm)

Quantiles from Model Simulated Data
precipitation Empirical Quantiles (mm)
- 1-1 line
- regression line
- 95% confidence bands

Density
Precipitation Quantiles (mm)
- Empirical
- Modeled

Return Level (mm)
Return Period (years)

**Fig. 5.** Tfit4: The Generalized Extreme Value (GEV) statistical distribution return period values, empirical and modeled fits with 95% confidence level using CMIP5 RCP2.6 precipitation (P, units = mm) during

[Figure]

**Fig. 6.** Tfit5: The Generalized Extreme Value (GEV) statistical distribution return period values, empirical and modeled fits with 95% confidence level using CRU precipitation (P, units = mm) during 1901-200